# RNA polymerase mutations cause cephalosporin resistance in clinical *Neisseria gonorrhoeae* isolates

**Samantha G Palace[1,2], Yi Wang[1], Daniel HF Rubin[1], Michael A Welsh[3], Tatum D Mortimer[1], Kevin Cole[4], David W Eyre[5], Suzanne Walker[3], Yonatan H Grad[1,2,6]\***

[1]Department of Immunology and Infectious Diseases, Harvard T. H. Chan School of Public Health, Boston, United States; [2]Center for Communicable Disease Dynamics, Harvard T. H. Chan School of Public Health, Boston, United States; [3]Department of Microbiology, Harvard Medical School, Boston, United States; [4]Public Health England, Royal Sussex County Hospital, Brighton, United Kingdom; [5]Big Data Institute, University of Oxford, Oxford, United Kingdom; [6]Division of Infectious Diseases, Brigham and Women's Hospital, Harvard Medical School, Boston, United States

**Abstract** Increasing *Neisseria gonorrhoeae* resistance to ceftriaxone, the last antibiotic recommended for empiric gonorrhea treatment, poses an urgent public health threat. However, the genetic basis of reduced susceptibility to ceftriaxone is not completely understood: while most ceftriaxone resistance in clinical isolates is caused by target site mutations in *penA*, some isolates lack these mutations. We show that *penA*-independent ceftriaxone resistance has evolved multiple times through distinct mutations in *rpoB* and *rpoD*. We identify five mutations in these genes that each increase resistance to ceftriaxone, including one mutation that arose independently in two lineages, and show that clinical isolates from multiple lineages are a single nucleotide change from ceftriaxone resistance. These RNA polymerase mutations cause large-scale transcriptional changes without altering susceptibility to other antibiotics, reducing growth rate, or deranging cell morphology. These results underscore the unexpected diversity of pathways to resistance and the importance of continued surveillance for novel resistance mutations.

*For correspondence:
ygrad@hsph.harvard.edu

**Competing interests:** The authors declare that no competing interests exist.

## Introduction

The rising incidence of *Neisseria gonorrhoeae* infection and antimicrobial resistance imperils effective therapy for gonococcal infections (*Centers for Disease Control and Prevention, 2018*; *Demczuk et al., 2015*; *Eyre et al., 2018*; *Kirkcaldy et al., 2016*). Current first-line therapy for gonorrhea relies on the extended spectrum cephalosporin ceftriaxone (CRO) as the backbone of treatment. Dual therapy with azithromycin was intended to delay the emergence of resistance, but rising azithromycin resistance rates have been reported (*Kirkcaldy et al., 2016*; *Public Health England, 2018*), resulting in revision of United Kingdom treatment guidelines to CRO monotherapy (*Fifer et al., 2019*). With no clear next-line agent, gonococcal resistance to ESCs is a problem of paramount clinical importance, illustrated by recent reports of treatment failure and global spread of a multidrug-resistant strain (*Eyre et al., 2018*; *Eyre et al., 2019*).

Rapid point-of-care diagnostics for antimicrobial susceptibility could help ensure efficacious treatment and improve stewardship (*Crofts et al., 2017*; *Tuite et al., 2017*), and genotypic testing for known resistance determinants has been implemented (*Allan-Blitz et al., 2017*; *Fifer et al., 2019*). In *N. gonorrhoeae*, reduced susceptibility to ceftriaxone (CRO[RS]) is most commonly associated with

genetic variation in *penA* (*Lindberg et al., 2007*; *Whiley et al., 2010*; *Zhao et al., 2009*). *penA* encodes PBP2, which is predicted to be the primary target of ESCs by homology to *E. coli* (*Kocaoglu and Carlson, 2015*); in agreement with this model, *penA* mutation and altered PBP2 affinity is the major mechanism of reduced ESC susceptibility in gonococci (*Dougherty et al., 1981*; *Ochiai et al., 2007*; *Tomberg et al., 2017*). Because characterized *penA* alleles correlate very strongly with phenotypic CRO$^{RS}$ in collections of clinical gonococcal isolates, the development of *penA*-targeted molecular diagnostic tests to rapidly evaluate gonococcal cephalosporin susceptibility is underway (*Deng et al., 2019*; *Zhao et al., 2019*).

However, these characterized collections often include isolates with phenotypic ESC$^{RS}$ that is not attributable to *penA* variation (*Demczuk et al., 2015*; *Grad et al., 2016*; *Peng et al., 2019*). For example, a recent report (*Abrams et al., 2017*) described the clinical gonococcal isolate GCGS1095, which has a cefixime (CFX) minimum inhibitory concentration (MIC) of 1 µg/mL and a CRO MIC of 0.5 µg/mL. These values are above the Center for Disease Control and Prevention's Gonococcal Isolate Surveillance Project (GISP) threshold values for CFX$^{RS}$ (0.25 µg/mL) and CRO$^{RS}$ (0.125 µg/mL) (*Kirkcaldy et al., 2016*). The type IX (non-mosaic) *penA* allele found in this isolate, also designated NG-STAR 9.001, is not known to contribute to reduced cephalosporin susceptibility and is found in many ESC susceptible isolates (*Whiley et al., 2007*).

Loci distinct from *penA* can also contribute to ESC$^{RS}$. These include mutations in *porB* that decrease drug permeability, mutations that enhance drug efflux by the MtrCDE pump, and non-mosaic mutations in penicillin-binding proteins (*Lindberg et al., 2007*; *Whiley et al., 2010*). Mutations in *pilQ* have arisen during in vitro selection for cephalosporin resistance, although these variants have not been observed in clinical isolates (*Johnson et al., 2014*). Considering these other resistance-associated loci in addition to *penA* has been proposed as a way to refine sequence-based prediction of cephalosporin susceptibility (*Eyre et al., 2017*). However, the genotype of GCGS1095 at these loci does not explain its high-level CRO$^{RS}$ phenotype (*Abrams et al., 2017*). Rapid genotypic diagnostic tests for known resistance mutations would therefore misclassify this strain as CRO susceptible. Defining the basis for ESC$^{RS}$ in this isolate and others like it is critical for the development of robust sequence-based diagnostics of antimicrobial susceptibility, for supporting public health genomic and phenotypic surveillance programs, and for understanding the biology underlying cephalosporin resistance.

Here, we employed an experimental transformation-based approach to identify the genetic basis of reduced susceptibility in three clinical isolates with high-level CRO$^{RS}$. We found that each of these isolates has unique mutations in the RNA polymerase holoenzyme (RNAP) that cause CRO$^{RS}$. One of these RNAP mutations is also present in a clinical isolate from the U.K., belonging to a genetically distinct clade, that has otherwise unexplained CRO$^{RS}$. We also identified an additional two RNAP mutations that arose de novo to cause CRO$^{RS}$ in vitro. Furthermore, introducing RNAP mutations into diverse clinical isolates from multiple lineages was sufficient to cause high-level phenotypic CRO$^{RS}$, underscoring the ability of isolates from circulating clades to develop CRO$^{RS}$ via a single mutation in RNA polymerase.

## Results

### A single missense mutation in *rpoB* is the genetic basis for reduced ceftriaxone susceptibility in the clinical isolate GCGS1095

To identify the genetic basis of reduced susceptibility in GCGS1095, we transformed a ceftriaxone susceptible (CRO$^{S}$) recipient strain with genomic DNA from GCGS1095. We reasoned that minimizing the genetic distance between the resistant donor and susceptible recipient would improve the likelihood of identifying the causative locus for CRO$^{RS}$ and would reduce potentially confounding effects of divergent genomic backgrounds. Consequently, we selected GCGS0457, a close phylogenetic neighbor of GCGS1095 (*Figure 1*), as the CRO$^{S}$ recipient for these experiments. Transformants were selected on CRO plates, and those that acquired the CRO$^{RS}$ phenotype were characterized by whole genome sequencing to determine which variants they had inherited from GCGS1095.

We recovered multiple CRO$^{RS}$ transformants. GCGS0457 did not develop spontaneous CRO$^{RS}$ during culture or when transformed with autologous GCGS0457 genomic DNA. Whole genome sequencing of the CRO$^{RS}$ transformants showed that each transformant inherited the same single

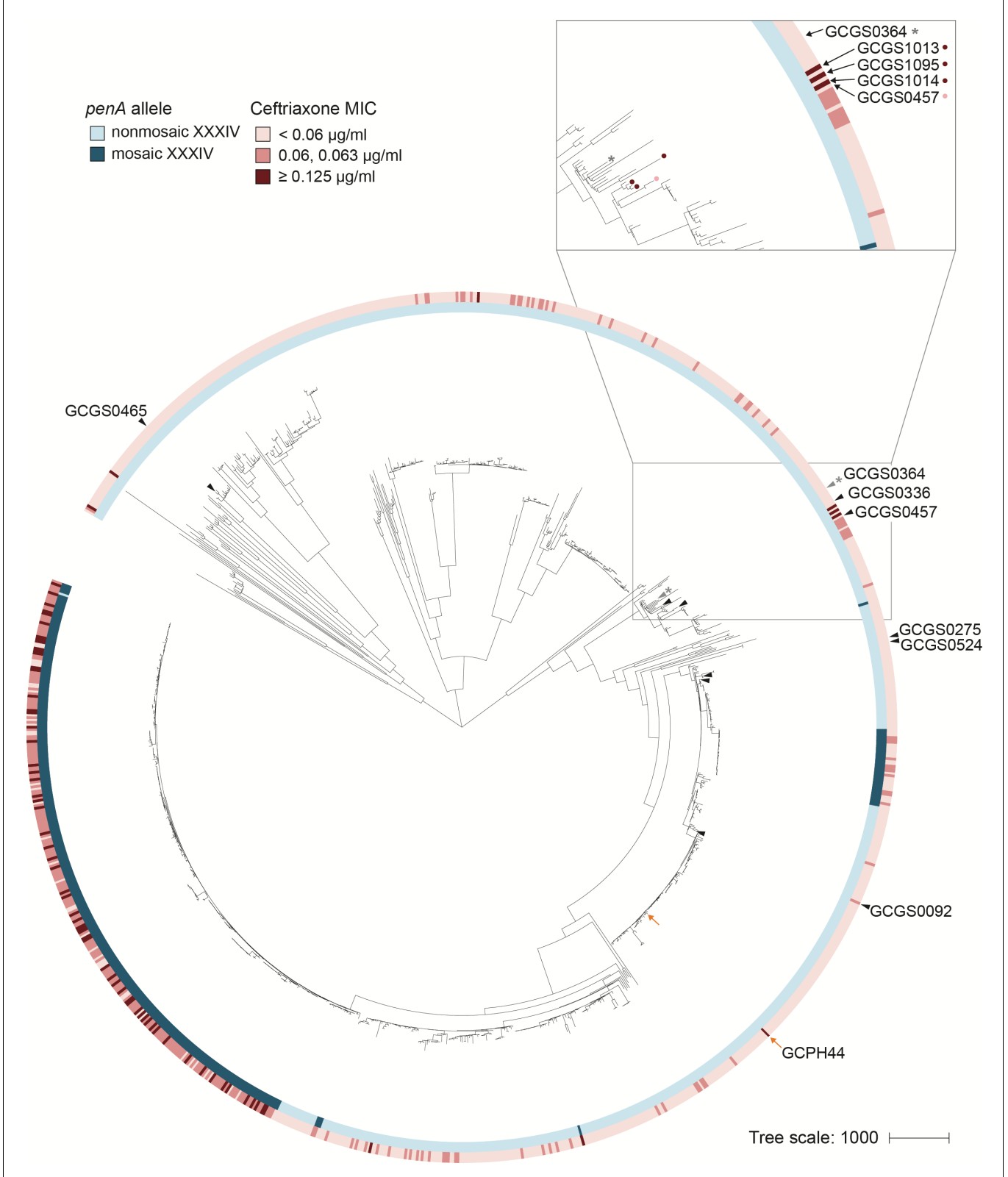

**Figure 1.** RNA polymerase-mediated reduced cephalosporin susceptibility among gonococcal isolates. Maximum likelihood phylogeny of the 1102 isolates of the GISP dataset and one United Kingdom isolate (GCPH44). Most high-level reduced cephalosporin susceptibility (MIC $\geq$0.125 µg/mL) in this dataset is associated with the mosaic *penA* XXXIV allele, but some isolates with high MICs lack this allele. In four of these isolates – GCGS1095, GCGS1014, GCGS1013 (inset, red-marked leaves) and the U.K. isolate GCPH44 (orange arrow) – CRO<sup>RS</sup> is caused by mutations in the RNA polymerase

*Figure 1 continued on next page*

*Figure 1 continued*

holoenzyme. Transformation of the CRO^RS allele *rpoB1* from GCGS1095 confers phenotypic reduced susceptibility to the phylogenetic neighbor GCGS0457 (inset, pink-marked leaf) and to other susceptible clinical isolates (black arrowheads). The isolate GCGS0364 (gray arrowhead, denoted with *) spontaneously develops CRO^RS via *rpoB* mutation in vitro.

nucleotide polymorphism (SNP) in *rpoB* (nucleotide change G602A), which encodes the RNA polymerase beta subunit RpoB (amino acid substitution R201H) (*Figure 2*). We named this allele *rpoB1*. These CRO^RS transformants did not acquire variants in genes known to contribute to reduced cephalosporin susceptibility, such as *penA*, *ponA*, *pilQ*, or *mtr* (*Eyre et al., 2017*).

As RNA polymerase mutations have not been associated with cephalosporin resistance in *Neisseria*, we tested whether the *rpoB1* mutation is sufficient to confer CRO^RS. We performed directed transformation of GCGS0457 with a ~ 1.5 kilobase PCR product surrounding the variant position of *rpoB1*. With the exception of the single *rpoB* G602A nucleotide change, the sequence of this PCR product was identical to the parental GCGS0457 sequence. Transformation with the *rpoB1* allele amplified from GCGS1095 yielded multiple CRO^RS transformants, whereas transformation with control PCR products amplified from a strain containing wild-type *rpoB* yielded none. Sanger sequencing confirmed the presence of the expected SNP in each resistant transformant. Representative CRO^RS transformants were further examined by whole genome sequencing, which ruled out spontaneous mutations. The CRO MIC of the *rpoB1* transformant strain GCGS0457 RpoB^R201H was 0.19 µg/mL, a more than ten-fold increase from recipient strain GCGS0457 (CRO MIC 0.012 µg/mL) (*Table 1*). This transformant MIC is similar to the MIC of the parental CRO^RS isolate GCGS1095 (*Table 1*), indicating that introduction of this SNP is sufficient to fully recapitulate the CRO^RS phenotype in the GCGS0457 background.

## RNA polymerase mutations explain CRO^RS in other clinical isolates

GCGS1014 is a phylogenetic neighbor of GCGS1095 that also has unexplained CRO^RS (*Figure 1*; *Table 1*). The close phylogenetic relationship between these isolates led us to hypothesize that they would share the same mechanism of CRO^RS. However, the GCGS1014 isolate lacks the *rpoB1* allele identified in GCGS1095. In fact, among the 1102 sequenced isolates in the GISP collection, the *rpoB1* allele is unique to GCGS1095 (*Supplementary file 1*).

To define the genetic basis of CRO^RS in GCGS1014, we used the same unbiased transformation approach as described above, again using the CRO^S neighbor GCGS0457 as the recipient strain.

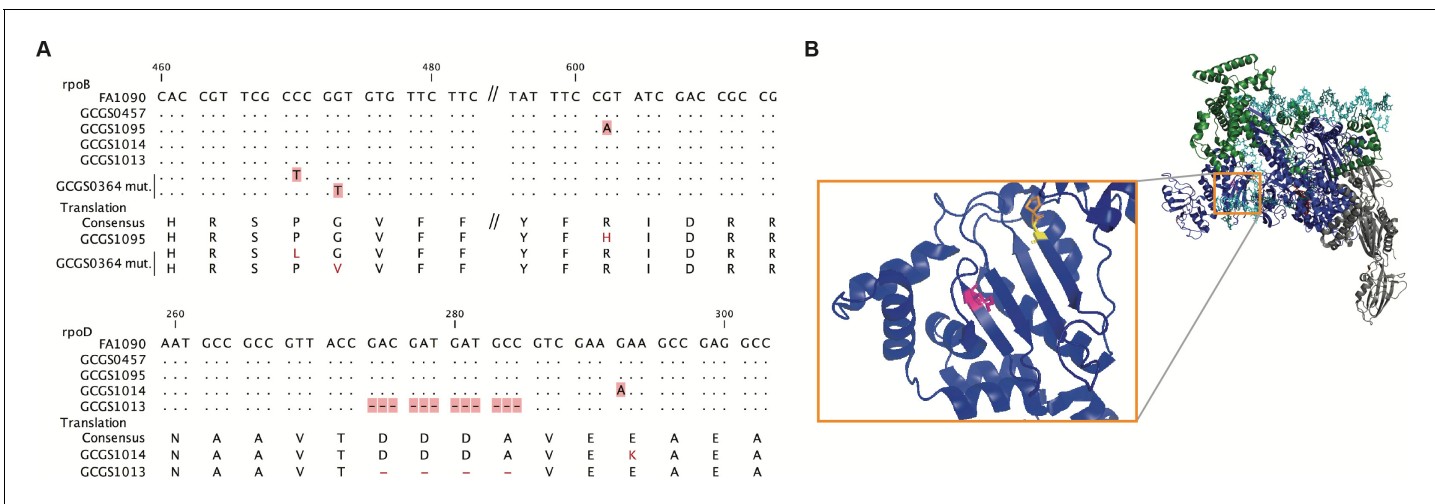

**Figure 2.** Location of CRO^RS-associated RNA polymerase mutations. (A) Alignment of mutant RNA polymerase alleles associated with reduced ceftriaxone susceptibility. (B) Crystal structure of the RNA polymerase holoenzyme from *E. coli* by Zuo et al. (PDB 4YLO) (*Zuo and Steitz, 2015*), showing the location of the residues homologous to *N. gonorrhoeae* RpoB R201 (magenta), G158 (yellow), and P157 (orange). The flexible region of σ^70 1.1 that includes the variant positions of *rpoD1* and *rpoD1* is not included in this structure, but is predicted to interact with this region of RpoB (*Murakami, 2013*).

**Table 1.** Selected strains, phenotypic ceftriaxone susceptibility, and relevant genotypes.

| strain | CRO MIC (µg/mL) | *penA* (PBP2) | *ponA* (PBP1) | PorB 120/121 | RpoB | RpoD |
|---|---|---|---|---|---|---|
| GCGS0457 | 0.012 | Type IX non-mosaic (NG STAR 9.001) | 421P | 120G/121V | wildtype | wildtype |
| GCGS1095 | 0.19 | Type IX non-mosaic (NG STAR 9.001) | 421P | 120N/121A | R201H (*rpoB1*) | wildtype |
| GCGS1014 | 0.125 | Type IX non-mosaic (NG STAR 9.001) | 421P | 120D/121A | wildtype | E98K (*rpoD1*) |
| GCGS1013 | 0.19 | Type IX non-mosaic (NG STAR 9.001) | 421P | 120G/121A | wildtype | Δ92–95 (*rpoD2*) |
| GCPH44 | 0.125 | Type II non-mosaic (NG STAR 2.001) | 421L | 120G/121A | H553N | Δ92–95 (*rpoD2*) |
| GCGS0457 *rpoB1* | 0.19 | Type IX non-mosaic (NG STAR 9.001) | 421P | 120G/121V | R201H (*rpoB1*) | wildtype |
| GCGS0457 *rpoD1* | 0.125 | Type IX non-mosaic (NG STAR 9.001) | 421P | 120G/121V | wildtype | E98K (*rpoD1*) |
| GCGS0457 *rpoD2* | 0.19 | Type IX non-mosaic (NG STAR 9.001) | 421P | 120G/121V | wildtype | Δ92–95 (*rpoD2*) |
| GCGS0364 | 0.023 | Type IX non-mosaic (NG STAR 9.001) | 421P | 120K/121D | wildtype | wildtype |
| GCGS0364 *rpoB2* | 0.5 | Type IX non-mosaic (NG STAR 9.001) | 421P | 120K/121D | G158V (*rpoB2*) | wildtype |
| GCGS0364 *rpoB3* | 0.75 | Type IX non-mosaic (NG STAR 9.001) | 421P | 120K/121D | P157L (*rpoB3*) | wildtype |

The CRO$^{RS}$ transformants from GCGS1014 gDNA did not have mutations in *rpoB* or in other characterized resistance-associated genes. Instead, all CRO$^{RS}$ transformants inherited a single nucleotide substitution in *rpoD* (nucleotide G292A), which encodes the major housekeeping sigma factor RpoD, or σ$^{70}$ (amino acid substitution E98K) (*Figure 2*). This allele, *rpoD1*, is present in the CRO$^{RS}$ donor GCGS1014 but not in GCGS1095 (*Table 1*) or in any other isolate from the GISP collection (*Supplementary file 1*). To test the ability of the *rpoD1* allele to confer CRO$^{RS}$ to a susceptible strain, we transformed GCGS0457 with a ~ 1.5 kilobase PCR product that includes the single variant position in *rpoD1*. As with the directed transformation of *rpoB1*, the *rpoD1* allele amplified from GCGS1014 transformed GCGS0457 to high-level CRO$^{RS}$, but control PCR products amplified from strains with wild-type *rpoD* did not. The presence of the *rpoD1* allele in resistant transformants was confirmed by Sanger sequencing and by whole genome sequencing, which also ruled out the possibility of spontaneous mutations causing the CRO$^{RS}$ phenotype. The *rpoD1* transformant strain GCGS0457 RpoD$^{E98K}$ had a CRO MIC of 0.125 µg/mL, the same as that of the parental CRO$^{RS}$ isolate GCGS1014 (*Table 1*).

As the CRO$^{RS}$ phenotypes in GCGS1095 and GCGS1014 are caused by distinct RNAP mutations, we examined alleles of RNAP components in each of the 1102 genomes in the GISP collection (*Grad et al., 2016*) to identify additional RNAP variants that might explain other cases of uncharacterized CRO$^{RS}$ (*Supplementary file 1*). We found that the isolate GCGS1013 has a 12-basepair inframe internal deletion in *rpoD*, resulting in the deletion of amino acids 92–95 (*Figure 2*); GCGS1013 has a CRO$^{RS}$ phenotype but does not have a CRO$^{RS}$-associated *penA* allele (CRO MIC 0.19 µg/mL; *Table 1*). Transformation of the GCGS1013 *rpoD* allele, *rpoD2*, into the susceptible recipient GCGS0457 increased CRO resistance to 0.19 µg/mL (*Table 1*). The *rpoD2* allele is also present in a clinical gonococcal isolate from a recent United Kingdom strain collection (*De Silva et al., 2016*). This isolate, GCPH44 (genome accession number SRR3360905), has a CRO$^{RS}$ phenotype (CRO MIC 0.125 µg/mL) and belongs to distinct phylogenetic lineage from that of GCGS1013 (*Figure 1*). Like GCGS1013, the CRO$^{RS}$ phenotype of GCPH44 is not attributable to variation in *penA* or other known resistance determinants (*Table 1*).

The amino acids deleted in the *rpoD2* allele (Δ92–95) are in a similar region of the RpoD protein as the single amino acid change in the *rpoD1* allele (E98K) (*Figure 2A*). This flexible portion of the σ$^{70}$ 1.1 domain is not included in published RNA polymerase holoenzyme crystal structures. Modeling of *E. coli* RNAP predicts that σ$^{70}$ 1.1 interacts with the DNA channel formed by the β subunit encoded by *rpoB* prior to open complex formation (*Murakami, 2013*). In *E. coli*, the RpoB residue homologous to the substituted amino acid in *rpoB1* (*E. coli* RpoB R197) is located on the surface of the β lobe structure that makes up this same channel (*Figure 2B*). Taken together, these results demonstrate that RNAP-mediated CRO$^{RS}$ has arisen multiple times in clinical isolates of *N. gonorrhoeae*, and that these different mutations affect a similar region of the RNAP holoenzyme. They may therefore act via a similar mechanism to effect phenotypic CRO$^{RS}$.

## Clinical isolates from multiple lineages can achieve high-level CRO^RS via a single nucleotide change in *rpoB*

The *rpoB1* and *rpoD1* alleles each differ from wild-type by a single nucleotide. The ability of these mutations to cause high-level CRO^RS when introduced into the susceptible strain GCGS0457 indicates that some gonococcal isolates are a single-step mutation from CRO^RS. Because three of the four clinical isolates we identified with RNAP-mediated CRO^RS are clustered in one part of the gonococcal phylogeny (*Figure 1*), we tested whether acquisition of CRO^RS via this pathway is limited to strains within this particular clade – which would suggest that one or more genetic background factors are required – or whether RNAP mutation represents a broadly accessible evolutionary trajectory to CRO^RS.

To address this question, we sought to transform genetically diverse recipient strains to CRO^RS with the *rpoB1* allele. In a panel of 17 clinical isolates from the GISP collection, we found five additional recipients dispersed throughout the phylogeny that acquired phenotypic CRO^RS following directed transformation with *rpoB1* (*Figure 1*; *Supplementary file 1*). The remaining 12 isolates, as well as the laboratory strain 28BL, did not readily produce CRO^RS *rpoB1* transformants after 1–2 transformation attempts. These results indicate that genetic background compatibility with this resistance mechanism is not limited to a single gonococcal lineage, to recipients with particular alleles of any RNA polymerase component (*Supplementary file 1*), or to recipients with variants in *ponA* or *mtrR* associated with β-lactam resistance (*Eyre et al., 2017*) (*Supplementary file 2*). As transformation efficiency in these diverse isolates is variable, the failure to recover CRO^RS transformants from any particular strain following a single transformation attempt may reflect low efficiency rather than an absolute genetic incompatibility with an RNAP-mediated resistance mechanism. The successful transformation of *rpoB1* into these five additional isolates therefore represents a lower bound on, rather than exact count of, the number of isolates in this panel that can develop resistance via this pathway.

## A clinical isolate develops high-level CRO^RS in vitro through spontaneous *rpoB* mutation

While screening isolates for genetic compatibility with *rpoB*-mediated CRO^RS in the directed transformation experiments described above, we tested isolates in parallel for spontaneous development of CRO^RS. We found that the CRO^S isolate GCGS0364, located in a similar part of the phylogeny as the resistant isolates GCGS1095, GCGS1014, and GCGS1013 (*Figure 1*), produced stably CRO^RS derivatives when exposed to CRO in vitro. Sanger sequencing of the *rpoB* locus in two of these derivatives identified two different spontaneous *rpoB* mutations: *rpoB2* (nucleotide change C470T, amino acid change P157L) and *rpoB3* (nucleotide change G473T, amino acid change G158V) (*Figure 2A*). These de novo RpoB mutations are predicted to occur in the same β-sheet as the *rpoB1* variant position (R201H) identified in GCGS1095 (*Figure 2B*). Directed transformation of *rpoB2* or *rpoB3* into the parental (CRO^S) GCGS0364 isolate conferred phenotypic CRO^RS, increasing the CRO MIC more than 20-fold from 0.023 μg/mL to 0.5–0.75 μg/mL (*Table 1*).

## RNAP-mediated CRO^RS acts via a cephalosporin-specific mechanism

Because optimal RNA polymerase functionality is required for bacterial fitness, we tested the hypothesis that these RNAP mutations are not specific cephalosporin resistance mutations, but instead cause a slow-growth or general stress response phenotype that nonspecifically increases resistance to diverse classes of antimicrobial compounds.

As expected, the *rpoB1*, *rpoD1*, and *rpoD2* alleles conferred reduced susceptibility to ceftriaxone as well as cefixime (another third-generation cephalosporin) when transformed into the CRO^S recipient GCGS0457. However, these alleles did not affect susceptibility to antimicrobial drugs that do not target the cell wall, including ciprofloxacin, tetracycline, azithromycin, and rifampicin. Surprisingly, these mutations also failed to confer resistance to the non-cephalosporin β-lactams penicillin and ertapenem. The apparent cephalosporin specificity of these RNAP mutations in the GCGS0457 background is consistent with the antibiotic susceptibility profiles of the three CRO^RS clinical isolates in which they were first identified (GCGS1095 for *rpoB1*, GCGS1014 for *rpoD1*, and GCGS1013 for *rpoD2*) (*Table 2*).

**Table 2.** Antibiotic susceptibility profiles of CRO[RS] strains with RNAP mutations.

| Strain | MIC (µg/ml) | | | | | | | |
|---|---|---|---|---|---|---|---|---|
| | CRO | CFX | PEN | EPM | AZI | TET | CIP | RIF |
| GCGS0457 | 0.012 | 0.016 | 2 | 0.016 | 0.25 | 1 | ≤0.015 | 0.125 |
| GCGS1095 | 0.19 | 1 | 2 | 0.032 | 0.5 | 1 | ≤0.015 | ≤0.0625 |
| GCGS1014 | 0.125 | 0.5 | 1.5 | 0.032 | 0.25 | 1 | ≤0.015 | ≤0.0625 |
| GCGS1013 | 0.19 | 0.5 | 1.5 | 0.023 | 0.5 | 1 | ≤0.015 | ≤0.0625 |
| GCGS0457 RpoB[R201H] | 0.19 | >0.5 | 2 | 0.047 | 0.25 | 1 | ≤0.015 | ≤0.0625 |
| GCGS0457 RpoD[E98K] | 0.125 | 0.5 | 1.5 | 0.023 | 0.25 | 1 | ≤0.015 | ≤0.0625 |
| GCGS0457 RpoD[Δ92-95] | 0.19 | 0.5 | 2 | 0.023 | 0.25 | 1 | ≤0.015 | ≤0.0625 |

GCGS0457 *rpoB1* and GCGS0457 *rpoD1* transformants were further examined for evidence of a slow growth phenotype that might increase antibiotic tolerance. When grown on GCB agar (Difco), neither of these transformants had a reduced growth rate compared to the parental GCGS0457 strain (*Figure 3A*). Furthermore, the *rpoB1* and *rpoD1* transformants had no gross defects in cell morphology by transmission electron microscopy (TEM) (*Figure 3B*), although transformants were slightly smaller than the parental GCGS0457 strain (*Figure 3—figure supplement 1*). Taken together, these results do not support the hypothesis that RNAP mutations reduce CRO susceptibility via a slow growth or nonspecific tolerance phenotype.

## CRO[RS]-associated RNAP mutations induce widespread transcriptional changes

To measure the functional effect of the *rpoB1* and *rpoD1* alleles on RNA polymerase, we characterized the transcriptional profiles of the CRO[RS] isolates GCGS1095 (*rpoB1*) and GCGS1014 (*rpoD1*) and the susceptible isolate GCGS0457 by RNA-seq. The transcriptional profiles of the CRO[RS] isolates were similar to one another but distinct from GCGS0457 (*Figure 4—figure supplement 1*), with 1337 and 1410 annotated ORFs significantly altered in abundance, respectively. The transcriptomic differences we observe may partly reflect the genetic distance between these isolates (*Wadsworth et al., 2019*), as the resistant strains GCGS1095 and GCGS1014 are more closely

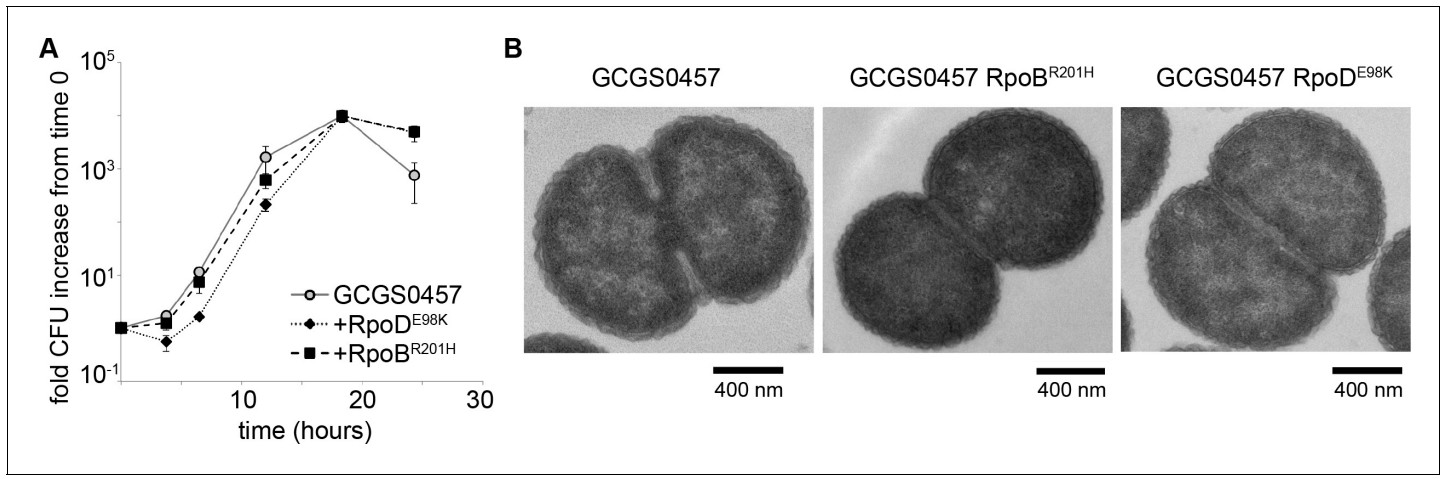

**Figure 3.** RNA polymerase mutations do not change growth phenotypes. (**A**) Growth of GCGS0457 and RNAP mutant transformants on solid GCB agar (mean and standard deviation of three technical replicates shown, representative of two independent experiments). RNAP mutations do not result in a growth rate defect. (**B**) Transmission electron micrographs of GCGS0457 and RNAP mutant transformants. CRO[RS] transformants are slightly smaller than the parental GCGS0457 strain, but are otherwise morphologically similar.

The online version of this article includes the following figure supplement(s) for figure 3:

**Figure supplement 1.** Size and eccentricity of GCGS0457 and CRO[RS] transformants.

related to one another than they are to GCGS0457. We therefore compared GCGS0457 to its isogenic CRO[RS] transformants GCGS0457 RpoB[R201H] and GCGS0457 RpoD[E98K]. The *rpoB1* and *rpoD1* alleles each profoundly altered the transcriptional profile of GCGS0457: 1278 annotated ORFs in GCGS0457 RpoD[E98K] and 1218 annotated ORFs in the GCGS0457 RpoB[R201H] were differentially expressed relative to the parental GCGS0457 strain. The transcriptional profiles of the two transformants were very similar to one another (*Figure 4—figure supplement 1*), indicating that these different mutations have similar functional consequences. Overall, a total of 890 annotated ORFs were differentially expressed in all four CRO[RS] strains (the two CRO[RS] isolates and two transformants in the GCGS0457 background) compared to GCGS0457. The vast majority of the transcriptional changes are small in magnitude: 805/890 (90%) are less than 2-fold differentially expressed in one or more of the CRO[RS] strains.

Altered expression of resistance-associated genes such as efflux pumps and β-lactamases is a common mechanism of transcriptionally-mediated drug resistance. Although these strains lack a TEM-1 β-lactamase (*Grad et al., 2016*), increased expression of the gonococcal MtrCDE efflux pump is known to increase resistance to β-lactams, including cephalosporins (*Lindberg et al., 2007*). However, transcription of the operon encoding this pump is not elevated in the CRO[RS] RNAP mutants (*Figure 4—figure supplement 2*), nor is it differentially upregulated in response to drug exposure in these strains (*Figure 4—figure supplement 3*).

In vitro evolution studies that evolved cephalosporin resistance in *N. gonorrhoeae* identified inactivating mutations in the type IV pilus pore subunit PilQ (*Johnson et al., 2014*; *Zhao et al., 2005*), although similar *pilQ* mutations have not been identified in clinical isolates, presumably because colonization requires pilus-mediated adhesion to epithelial cells (*Kellogg et al., 1963*; *Rudel et al., 1992*). Notably, *pilQ* transcription is reduced in the GCGS0457 *rpoB1* and GCGS0457 *rpoD1* transformants, as well as in the CRO[RS] isolates GCGS1095 and GCGS1014 (*Figure 4—figure supplements 2* and *3*). The effect of hypomorphic *pilQ* expression on cephalosporin susceptibility has not been described, but it is possible that reducing *pilQ* transcript levels may confer the benefits of inactivating *pilQ* mutations without altogether eliminating pilus-mediated attachment and colonization.

## CRO[RS]-associated RNAP mutations alter the abundance of cell wall biosynthesis enzymes

Since ESCs, like other β-lactams, block cell wall biosynthesis by covalently inhibiting penicillin-binding proteins (PBPs), we examined how CRO[RS]-associated RNAP mutations affect expression of enzymes in this target pathway. *N. gonorrhoeae* encodes four known PBPs (*Obergfell et al., 2018*; *Sauvage et al., 2008*): the essential high-molecular weight bifunctional transpeptidase/transglycosylase PBP1 (encoded by *ponA*); the essential high-molecular weight transpeptidase PBP2 (encoded by *penA*) (*Zhao et al., 2009*), which is the primary target of ESCs; and the nonessential carboxypeptidases PBP3 (encoded by *dacB*) and PBP4 (encoded by *pbpG*). In addition to these canonical PBPs, *N. gonorrhoeae* genomes include DacC (encoded by *dacC*), a third putative low-molecular weight carboxypeptidase that lacks conserved active site motifs (*Obergfell et al., 2018*). A homologue of the L,D-transpeptidase YnhG is also present in the genome, although it has not been reported to be functional in *N. gonorrhoeae*.

Increased expression of drug targets can lead to decreased drug susceptibility, but CRO[RS]-associated RNAP mutations did not increase mRNA or protein abundance of the CRO target PBP2 (*Figure 4*). Transcription of the putative L,D-transpeptidase gene *ynhG* is also unchanged in the CRO[RS] strains (*Figure 4—figure supplement 2*). By contrast, PBP1 (*ponA*) is upregulated in each of the CRO[RS] strains compared to GCGS0457, while expression of PBP3, *pbpG*, and *dacC* is decreased in these strains (*Figure 4A–B*). Similar expression patterns were observed in the presence of sub-inhibitory CRO (*Figure 4—figure supplement 3*).

These altered PBP expression patterns are reflected in the cell wall structure of the CRO[RS] strains: reduced expression of the carboxypeptidases results in more peptidoglycan with pentapeptide stems (*Figure 5* and *Supplementary file 3*), in agreement with reported structural changes of cell walls from strains lacking the DacB carboxypeptidase (*Obergfell et al., 2018*). The increased expression of PBP1 in these RNAP mutants does not appear to increase the abundance of crosslinked peptidoglycan (*Figure 5* and *Supplementary file 3*).

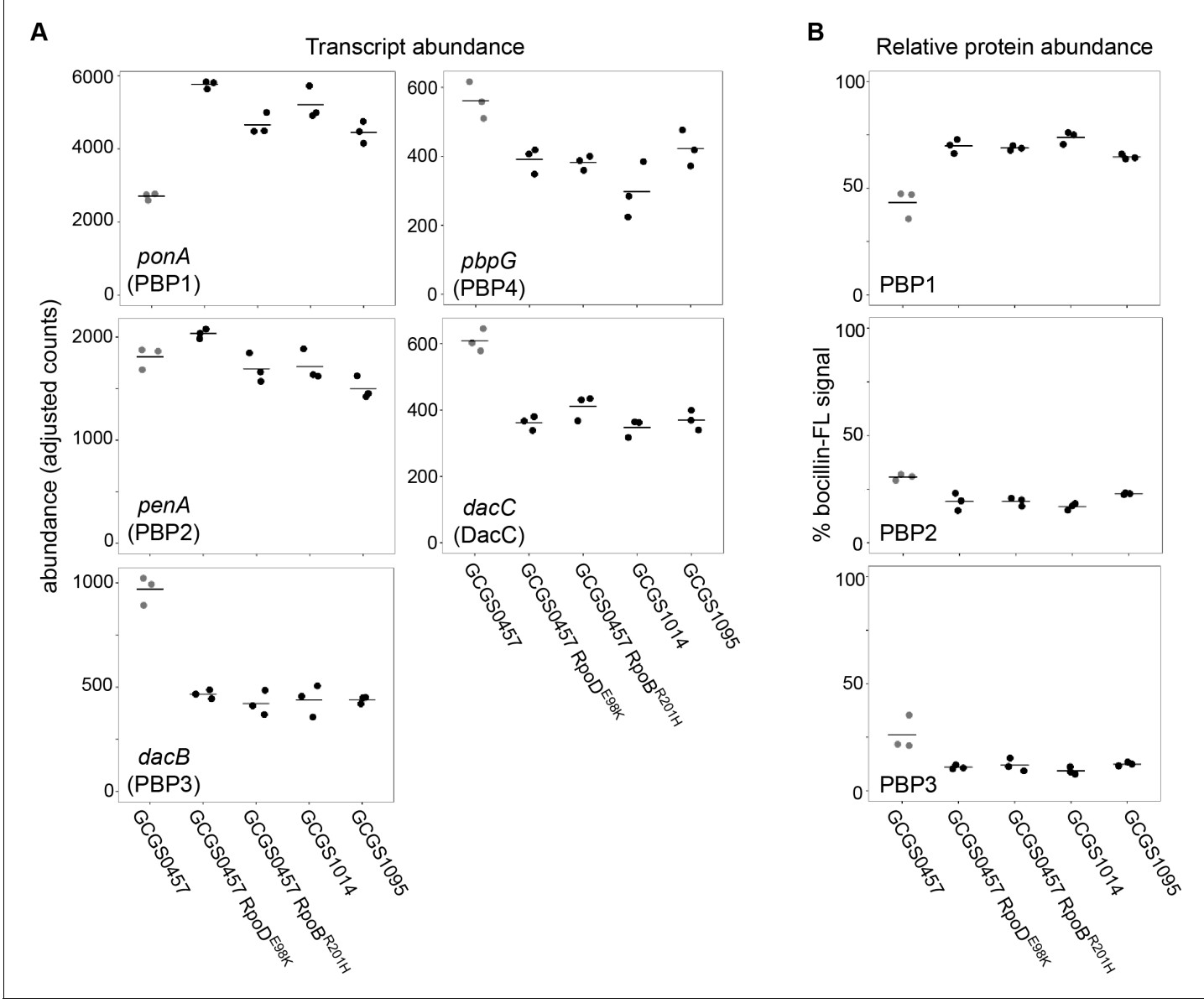

**Figure 4.** Effect of CRO^RS-associated RNA polymerase mutations on PBP abundance. (**A**) Normalized abundance of transcripts encoding penicillin binding proteins (PBPs) in parental isolates (GCGS0457, GCGS1095, GCGS1014) and RNAP mutant transformants in the GCGS0457 background, measured by RNAseq. (**B**) Relative protein abundance of PBP1, PBP2, and PBP3, as measured by bocillin-FL labeling. Total bocillin-FL signal for each strain was set at 100%. The relative contribution of each PBP to that signal is shown here. PBP4 protein was not observed on these gels, in agreement with previous reports (*Stefanova et al., 2003*).

The online version of this article includes the following figure supplement(s) for figure 4:

**Figure supplement 1.** Principle components analysis of RNA-seq data.

**Figure supplement 2.** Transcript abundance of various genes in CRO^RS strains with RNA polymerase mutations.

**Figure supplement 3.** Transcript abundance of various genes in CRO^RS strains with RNAP mutations, exposed to sub-inhibitory ceftriaxone.

## Discussion

We have identified five different CRO^RS-associated RNA polymerase alleles. Three of these – *rpoB1*, *rpoD1*, and *rpoD2* – are the genetic basis of high-level reduced ESC susceptibility in clinical isolates with previously unexplained resistance phenotypes. The remaining two arose spontaneously during culture, indicating that the RNAP alleles described in this work are likely only a subset of functionally similar mutations that could arise in clinical isolates. Clinical isolates from diverse genetic

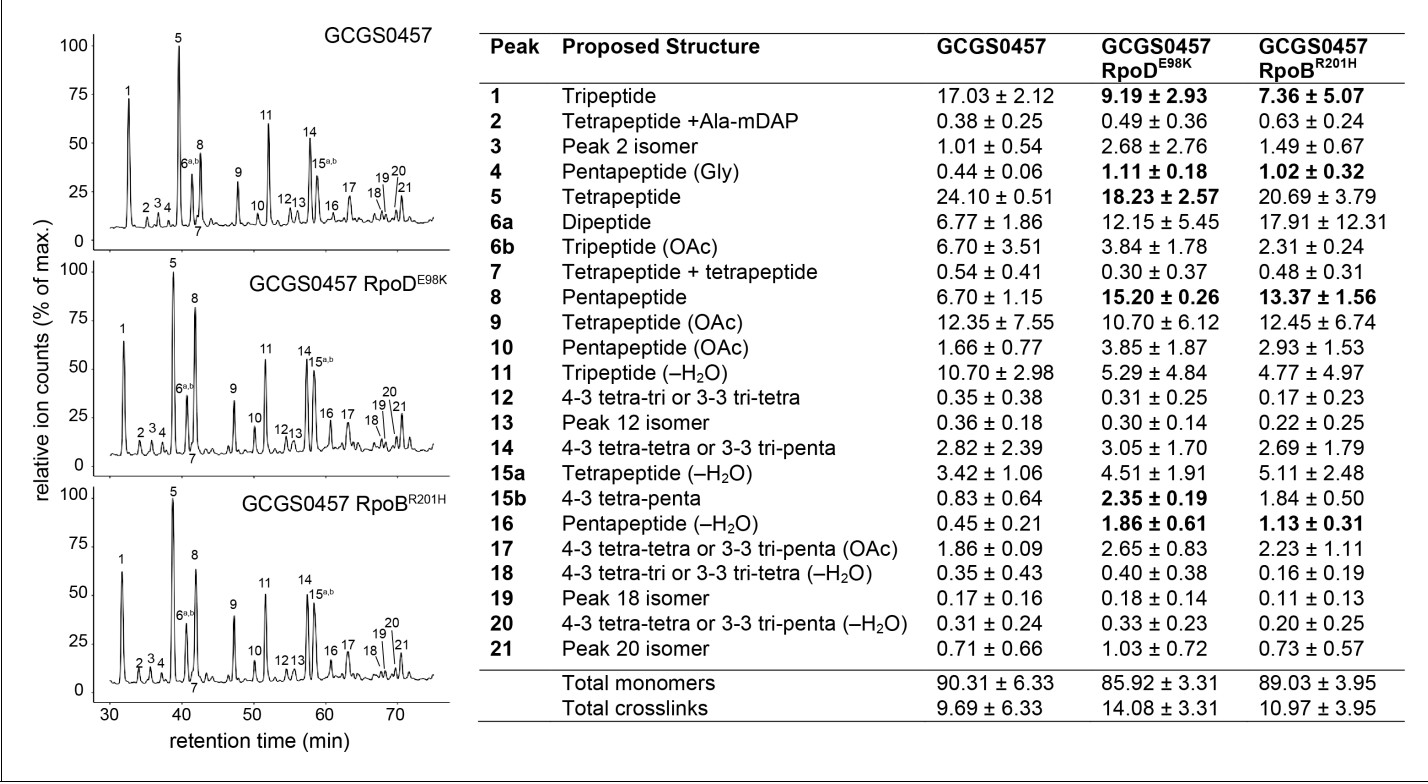

| Peak | Proposed Structure | GCGS0457 | GCGS0457 RpoD[E98K] | GCGS0457 RpoB[R201H] |
|---|---|---|---|---|
| 1 | Tripeptide | 17.03 ± 2.12 | **9.19 ± 2.93** | **7.36 ± 5.07** |
| 2 | Tetrapeptide +Ala-mDAP | 0.38 ± 0.25 | 0.49 ± 0.36 | 0.63 ± 0.24 |
| 3 | Peak 2 isomer | 1.01 ± 0.54 | 2.68 ± 2.76 | 1.49 ± 0.67 |
| 4 | Pentapeptide (Gly) | 0.44 ± 0.06 | **1.11 ± 0.18** | **1.02 ± 0.32** |
| 5 | Tetrapeptide | 24.10 ± 0.51 | **18.23 ± 2.57** | 20.69 ± 3.79 |
| 6a | Dipeptide | 6.77 ± 1.86 | 12.15 ± 5.45 | 17.91 ± 12.31 |
| 6b | Tripeptide (OAc) | 6.70 ± 3.51 | 3.84 ± 1.78 | 2.31 ± 0.24 |
| 7 | Tetrapeptide + tetrapeptide | 0.54 ± 0.41 | 0.30 ± 0.37 | 0.48 ± 0.31 |
| 8 | Pentapeptide | 6.70 ± 1.15 | **15.20 ± 0.26** | **13.37 ± 1.56** |
| 9 | Tetrapeptide (OAc) | 12.35 ± 7.55 | 10.70 ± 6.12 | 12.45 ± 6.74 |
| 10 | Pentapeptide (OAc) | 1.66 ± 0.77 | 3.85 ± 1.87 | 2.93 ± 1.53 |
| 11 | Tripeptide (−$H_2O$) | 10.70 ± 2.98 | 5.29 ± 4.84 | 4.77 ± 4.97 |
| 12 | 4-3 tetra-tri or 3-3 tri-tetra | 0.35 ± 0.38 | 0.31 ± 0.25 | 0.17 ± 0.23 |
| 13 | Peak 12 isomer | 0.36 ± 0.18 | 0.30 ± 0.14 | 0.22 ± 0.25 |
| 14 | 4-3 tetra-tetra or 3-3 tri-penta | 2.82 ± 2.39 | 3.05 ± 1.70 | 2.69 ± 1.79 |
| 15a | Tetrapeptide (−$H_2O$) | 3.42 ± 1.06 | 4.51 ± 1.91 | 5.11 ± 2.48 |
| 15b | 4-3 tetra-penta | 0.83 ± 0.64 | **2.35 ± 0.19** | 1.84 ± 0.50 |
| 16 | Pentapeptide (−$H_2O$) | 0.45 ± 0.21 | **1.86 ± 0.61** | **1.13 ± 0.31** |
| 17 | 4-3 tetra-tetra or 3-3 tri-penta (OAc) | 1.86 ± 0.09 | 2.65 ± 0.83 | 2.23 ± 1.11 |
| 18 | 4-3 tetra-tri or 3-3 tri-tetra (−$H_2O$) | 0.35 ± 0.43 | 0.40 ± 0.38 | 0.16 ± 0.19 |
| 19 | Peak 18 isomer | 0.17 ± 0.16 | 0.18 ± 0.14 | 0.11 ± 0.13 |
| 20 | 4-3 tetra-tetra or 3-3 tri-penta (−$H_2O$) | 0.31 ± 0.24 | 0.33 ± 0.23 | 0.20 ± 0.25 |
| 21 | Peak 20 isomer | 0.71 ± 0.66 | 1.03 ± 0.72 | 0.73 ± 0.57 |
| | Total monomers | 90.31 ± 6.33 | 85.92 ± 3.31 | 89.03 ± 3.95 |
| | Total crosslinks | 9.69 ± 6.33 | 14.08 ± 3.31 | 10.97 ± 3.95 |

**Figure 5.** Effect of CRO[RS]-associated RNA polymerase mutations on cell wall structure. Relative abundance of muropeptide peaks in cell wall digests of GCGS0457 and its CRO[RS] derivatives, GCGS0457 RpoD[E98K] and GCGS0457 RpoB[R201H]. Transformant cell walls contain a higher proportion of peptidoglycan with pentapeptide stems (peaks 4, 8, 15b, and 16). Relative abundance values for each muropeptide species (right) were calculated by extracting the peak mass from the total ion chromatogram, integrating the resulting peak area, and dividing by the sum of all of the peak areas within each sample. Shown: mean and standard deviation from 3 biological replicates (data from individual replicates can be found in *Figure 5—source data 1*). Muropeptide species that are significantly altered in abundance each transformant compared to the parental strain GCGS0457 are indicated in bold ($p < 0.05$ by two-tailed t test). See *Supplementary file 3* for a list of all muropeptide masses detected.

The online version of this article includes the following source data for figure 5:

**Source data 1.** Relative abundance of muropeptide species in individual biological replicates.

backgrounds can achieve high-level CRO[RS] via these RNAP mutations. This is highlighted by the appearance of the *rpoD2* allele in a second clinical isolate belonging to a genetically distinct clade. Because this isolate is a member of a large clade that includes many closely related isolates, this result illustrates the concerning potential of these mutations to cause *penA*-independent CRO[RS] in globally distributed strains and the importance of including these mutations in ongoing surveillance efforts.

Among the isolates of the GISP collection, there was no evidence for sustained transmission of the CRO[RS] RNAP mutant isolates (*Figure 1*), perhaps indicating that fitness costs associated with these RNAP variants have prevented their spread. However, it is important to note that the CRO[RS] isolates GCGS1095, GCGS1014, and GCGS1013 are susceptible to azithromycin (*Table 2*). This susceptibility may be an alternative explanation for the failure of these RNAP mutations to spread in an era of azithromycin/ceftriaxone combination therapy.

RNA polymerase mutations have been identified as mediators of diverse phenotypes, such as reduced susceptibility to phage lysis (*Atkinson and Gottesman, 1992*; *Obuchowski et al., 1997*) and antimicrobial drugs, including cell wall biosynthesis inhibitors (*Cui et al., 2010*; *Kristich and Little, 2012*; *Lee et al., 2013*; *Penwell et al., 2015*; *Wang et al., 2017*). These mutations often arise during in vitro evolution experiments, and are typically described as artifacts of in vitro conditions, as the presumed pleiotropy of such mutations is typically thought to be prohibitively deleterious. This description of resistance-associated RNAP mutations in isolates collected from symptomatic patients demonstrates that, at least in the case of CRO[RS] in *N. gonorrhoeae*, clinical isolates can

acquire resistance via RNAP mutation while still maintaining sufficient fitness to cause disease. Similar results have been reported regarding the role of *rpoB* mutation in decreased susceptibility to vancomycin in staphylococcal species (*Guérillot et al., 2018*; *Lee et al., 2018*; *Watanabe et al., 2011*).

In accordance with this model, the RNAP mutations identified in these CRO[RS] isolates result in an apparently narrowly-targeted phenotypic change: RNAP mutants acquire resistance to cephalosporins but not other classes of antimicrobial drugs, show no growth defect in vitro, and display broad but small-magnitude changes to transcript levels, indicating that RNAP mutation may be a valid approach to fine-tuning bacterial physiology for enhanced survival under certain adverse conditions, such as antibiotic exposure.

The observation that the RNAP mutations identified here neither decrease susceptibility to other antimicrobials nor alter growth phenotypes supports the hypothesis that these mutations generate a 'fine-tuning' cephalosporin-specific resistance mechanism, as opposed to a large-scale physiological shift toward a generally antibiotic-tolerant state. Altered activity of the RNA polymerase transcription machinery likely results in this type of drug-specific reduced susceptibility. RNAP variants differentially express multiple transcripts relating to the cephalosporin mode of action. For example, while PBP2 levels are unaffected, the increase of PBP1 expression may contribute to cephalosporin resistance. PBP1 and PBP2 both catalyze transpeptidation, but PBP1 belongs to a class of enzymes that is not efficiently inhibited by third-generation cephalosporins (*Kocaoglu and Carlson, 2015*), suggesting that increased PBP1 levels may partly compensate for CRO-inhibited PBP2 during drug treatment. The RNAP variants simultaneously reduce D,D-carboxypeptidase expression, increasing the pool of pentapeptide peptidoglycan monomers available for transpeptidation by PBP1 and PBP2. Decreased expression of pilus pore components in these mutants may also contribute to CRO[RS] by reducing permeability of the outer membrane to cephalosporins (*Chen et al., 2004*; *Johnson et al., 2014*; *Nandi et al., 2015*; *Zhao et al., 2005*). These expression changes and others may work additively or synergistically to increase drug resistance. The pleiotropy of RNAP mutations may thus enable a multicomponent resistance mechanism to emerge from a single genetic change.

This result has important implications regarding the biology underlying cephalosporin resistance, the potential for de novo evolution of resistance under cephalosporin monotherapy, and the accuracy of sequence-based diagnostics. The identification of five independent mutations in two separate components of the RNA polymerase machinery illustrates challenges faced by computational genomics strategies to define new resistance alleles, especially because there are other variants in these genes – such as rifampicin resistance mutations in *rpoB* – that do not confer CRO[RS]. Continued isolate collection, phenotypic characterization, and traditional genetic techniques will be critical for defining emerging resistance mechanisms (*Hicks et al., 2019*). The observation that multiple lineages can gain CRO resistance through RNAP mutations further underscores the need to monitor for the development of CRO[RS] through pathways other than *penA* mutation, particularly as the ESCs are increasingly relied upon for treatment. Identifying these alternative genetic mechanisms of reduced susceptibility is needed to support the development of accurate and reliable sequence-based diagnostics that predict CRO susceptibility, as well as to aid in the design of novel therapeutics.

## Materials and methods

### Key resources table

| Reagent type (species) or resource | Designation | Source or reference | Identifiers | Additional information |
|---|---|---|---|---|
| Gene (*Neisseria gonorrhoeae*) | *penA* | *Neisseria gonorrhoeae* PubMLST | NEIS1753 | |
| Gene (*Neisseria gonorrhoeae*) | *ponA* | *Neisseria gonorrhoeae* PubMLST | NEIS0414 | |

*Continued on next page*

*Continued*

| Reagent type (species) or resource | Designation | Source or reference | Identifiers | Additional information |
|---|---|---|---|---|
| Gene (*Neisseria gonorrhoeae*) | *rpoB* | *Neisseria gonorrhoeae* PubMLST | NEIS0123 | |
| Gene (*Neisseria gonorrhoeae*) | *rpoD* | *Neisseria gonorrhoeae* PubMLST | NEIS1466 | |
| Strain, strain background (*Neisseria gonorrhoeae*) | GCGS0457 | (*Grad et al., 2016*) | | |
| Strain, strain background (*Neisseria gonorrhoeae*) | GCGS1095 | (*Grad et al., 2016*) | | |
| Strain, strain background (*Neisseria gonorrhoeae*) | GCGS1014 | (*Grad et al., 2016*) | | |
| Strain, strain background (*Neisseria gonorrhoeae*) | GCGS1013 | (*Grad et al., 2016*) | | |
| Strain, strain background (*Neisseria gonorrhoeae*) | GCGS0364 | (*Grad et al., 2016*) | | |
| Strain, strain background (*Neisseria gonorrhoeae*) | GCGS0092 | (*Grad et al., 2016*) | | |
| Strain, strain background (*Neisseria gonorrhoeae*) | GCGS0126 | (*Grad et al., 2016*) | | |
| Strain, strain background (*Neisseria gonorrhoeae*) | GCGS0161 | (*Grad et al., 2016*) | | |
| Strain, strain background (*Neisseria gonorrhoeae*) | GCGS0249 | (*Grad et al., 2016*) | | |
| Strain, strain background (*Neisseria gonorrhoeae*) | GCGS0275 | (*Grad et al., 2016*) | | |
| Strain, strain background (*Neisseria gonorrhoeae*) | GCGS0336 | (*Grad et al., 2016*) | | |
| Strain, strain background (*Neisseria gonorrhoeae*) | GCGS0353 | (*Grad et al., 2016*) | | |
| Strain, strain background (*Neisseria gonorrhoeae*) | GCGS0374 | (*Grad et al., 2016*) | | |

*Continued*

| Reagent type (species) or resource | Designation | Source or reference | Identifiers | Additional information |
|---|---|---|---|---|
| Strain, strain background (*Neisseria gonorrhoeae*) | GCGS0465 | (*Grad et al., 2016*) | | |
| Strain, strain background (*Neisseria gonorrhoeae*) | GCGS0489 | (*Grad et al., 2016*) | | |
| Strain, strain background (*Neisseria gonorrhoeae*) | GCGS0497 | (*Grad et al., 2016*) | | |
| Strain, strain background (*Neisseria gonorrhoeae*) | GCGS0501 | (*Grad et al., 2016*) | | |
| Strain, strain background (*Neisseria gonorrhoeae*) | GCGS0524 | (*Grad et al., 2016*) | | |
| Strain, strain background (*Neisseria gonorrhoeae*) | GCGS0560 | (*Grad et al., 2016*) | | |
| Strain, strain background (*Neisseria gonorrhoeae*) | GCGS0576 | (*Grad et al., 2016*) | | |
| Strain, strain background (*Neisseria gonorrhoeae*) | GCGS0698 | (*Grad et al., 2016*) | | |
| Strain, strain background (*Neisseria gonorrhoeae*) | GCGS0769 | (*Grad et al., 2016*) | | |
| Strain, strain background (*Neisseria gonorrhoeae*) | GCPH44 | (*De Silva et al., 2016*) | | |
| Strain, strain background (*Neisseria gonorrhoeae*) | 28BL | Gift of S. Johnson | | |
| Chemical compound, drug | Ceftriaxone disodium salt hemi (heptahydrate) | Sigma Aldrich | Cat. # C5793 | |
| Chemical compound, drug | Ceftriaxone Etest | bioMérieux | SKU # 412302 and 412300 | |
| Chemical compound, drug | Penicillin G Etest | bioMérieux | SKU # 412262 | |
| Chemical compound, drug | Ertapenem Etest | bioMérieux | SKU # 531640 | |

*Continued on next page*

*Continued*

| Reagent type (species) or resource | Designation | Source or reference | Identifiers | Additional information |
|---|---|---|---|---|
| Chemical compound, drug | Cefixime trihydrate | Sigma Aldrich | Cat. # 18588 | |
| Chemical compound, drug | Azithromycin | Sigma Aldrich | Cat. # PHR1088 | |
| Chemical compound, drug | Tetracycline hydrochloride | Sigma Aldrich | Cat. # T7660 | |
| Chemical compound, drug | Ciprofloxacin | Sigma Aldrich | Cat. # 17850 | |
| Chemical compound, drug | Rifampicin | Chem-Impex | Cat. # 00260 | |
| Chemical compound, drug | Bocillin FL penicillin, sodium salt | Thermo Fisher | Cat. # B13233 | |
| Sequence-based reagent | RpoB-US | This paper | PCR primers | ATGCCGTCTGAAT ATCAGATTGATGC GTACCGTT |
| Sequenced-based reagent | RpoB-DS | This paper | PCR primers | CGTACTCGACGGT TGCCCAAG |
| Sequenced-based reagent | RpoD-US | This paper | PCR primers | AACTGCTCGGACA GGAAGCG |
| Sequenced-based reagent | RpoD-DS | This paper | PCR primers | CGCGTTCGAGTTT GCGGATGTT |

## Bacterial strains and culture conditions

Strains are presented in *Supplementary file 4*. Metadata for the 1102 GISP isolates included in *Figure 1* was previously published (*Grad et al., 2016*). Except where otherwise specified, *N. gonorrhoeae* was cultured on GCB agar (Difco) with Kellogg's supplements (GCB-K) (*Kellogg et al., 1963*) at 37°C in a 5% $CO_2$ atmosphere. Antibiotic susceptibility testing of *N. gonorrhoeae* strains was performed on GCB media supplemented with 1% IsoVitaleX (Becton Dickinson) via agar dilution (for cefixime, azithromycin, tetracycline, ciprofloxacin, and rifampicin) or Etest (bioMérieux) (for ceftriaxone, penicillin G, and ertapenem). A $CRO^{RS}$ breakpoint of $\geq 0.125$ µg/mL was used, in accordance with the GISP breakpoint (*Kirkcaldy et al., 2016*).

## Transformation of reduced cephalosporin susceptibility with genomic DNA

The $CRO^S$ recipient strain GCGS0457 was transformed with genomic DNA from GCGS1014 and GCGS1095. Transformations were conducted in liquid culture as described (*Morse et al., 1986*; *Wadsworth et al., 2018*). Briefly, piliated *N. gonorrhoeae* was suspended in GCP medium (15 g/L protease peptone 3, 1 g/L soluble starch, 4 g/L dibasic $K_2HPO_4$, 1 g/L monobasic $KH_2PO_4$, 5 g/L NaCl) with Kellogg's supplement, 10 mM $MgCl_2$, and approximately 100 ng genomic DNA. Suspensions were incubated for 10 min at 37°C with 5% $CO_2$. Transformants were allowed to recover on nonselective agar for 4–5 hr. After recovery, transformants were plated on GCB-K ceftriaxone gradient agar plates, which were prepared by allowing a 40 µL droplet of 5 µg/mL ceftriaxone to dry onto a GCB-K agar plate. Transformations performed with GCGS0457 genomic DNA served as controls to monitor for spontaneous $CRO^{RS}$ mutation. After outgrowth at 37°C in 5% $CO_2$, colonies growing within the ceftriaxone zone of inhibition were subcultured for further analysis.

## Transformation of RNA polymerase mutations

~1.5 kb PCR fragments surrounding the *rpoB* and *rpoD* loci of interest were amplified using the primer pairs RpoB-US (5'-ATGCCGTCTGAATATCAGATTGATGCGTACCGTT-3') and RpoB-DS (5'-CGTACTCGACGGTTGCCCAAG-3') or RpoD-US (5'-AACTGCTCGGACAGGAAGCG-3') and RpoD-DS (5'-CGCGTTCGAGTTTGCGGATGTT-3'). The 12 bp DNA uptake sequence (DUS) for *N. gonorrhoeae* (*Ambur et al., 2007*) was added to the 5' end of the RpoB-US primer (underlined) to enhance transformation efficiency with the PCR product; a DUS was not added to the RpoD-US or RpoD-DS primers, as the PCR product amplified by these primers includes two endogenous DUS regions. CRO^S recipient strains were transformed with 200–300 ng purified PCR products and transformants were selected for CRO^RS as above. Transformation reactions performed with wildtype alleles (*rpoB*^+ from GCGS1014; *rpoD*^+ from GCGS1095) served as controls to monitor for spontaneous CRO^RS mutation.

The following isolates were used for experiments examining the compatibility of *rpoB1*-mediated CRO^RS with diverse genetic backgrounds: GCGS0092, GCGS0275, GCGS0336, GCGS0465, GCGS0524 (all successfully); and GCGS0126, GCGS0161, GCGS0249, GCGS0353, GCGS0374, GCGS0489, GCGS0497, GCGS0501, GCGS0560, GCGS0576, GCGS0698, and GCGS0769. CRO susceptibility and RNA polymerase allele information for each of these isolates is available in *Supplementary file 1*.

## Sequencing and analysis

Following undirected transformation of GCGS0457 with GCGS1014 or GCGS1095 genomic DNA, or directed transformation of GCGS0457 with PCR products, CRO^RS transformants were analyzed by whole genome sequencing. Genomic DNA of transformants was purified with the PureLink Genomic DNA Mini kit (Life Technologies) and sequencing libraries were prepared as described *Kim et al. (2017)*. Paired-end sequencing of these libraries was performed on an Illumina MiniSeq (Illumina). Reads were aligned to the de novo assembly of the GCGS0457 genome (European Nucleotide Archive, accession number ERR855051) using bwa v0.7.8 (*Li, 2013*), and variant calling was performed with pilon v1.22 (*Walker et al., 2014*).

## Whole genome assembly and annotation

Whole genome sequencing reads from the GISP collection were assembled using Velvet (*Zerbino and Birney, 2008*) as previously described *Grad et al. (2016)*. Spades v 3.12 (*Bankevich et al., 2012*) was used for de novo assembly of GCPH44. Contigs were corrected by mapping reads back to the assembly (–careful), and contigs with less than 10X coverage or fewer than 500 nucleotides were removed. Reads were also mapped to the reference genome NCCP11945 (NC_011035.1) with bwa mem v0.7.15 (*Li, 2013*). Variants were identified with Pilon v 1.16 (*Walker et al., 2014*) using a minimum depth of 10 and minimum mapping quality of 20. SNPs, small deletions, and uncertain positions were incorporated into the reference genome to create pseudogenomes. For each isolate in the GISP collection and in the United Kingdom collection (*De Silva et al., 2016*), the sequence of RNA polymerase components was identified using BLASTn (ncbi-blast v2.2.30) (*Supplementary file 1*).

## Phylogenetic analysis

Recombinant regions of the pseudogenome alignment were detected by Gubbins (*Croucher et al., 2015*), and the maximum likelihood phylogenetic tree was estimated from this masked alignment using RAxML (*Stamatakis, 2014*). The phylogeny was visualized using ITOL (*Letunic and Bork, 2019*).

## Growth curves

Bacterial cells grown overnight on GCB-K plates were suspended in tryptic soy broth to $OD_{600}$ 0.01. Three replicate GCB-K plates per time point were inoculated with 0.1 mL of this suspension and incubated at 37°C with 5% $CO_2$. At each time point, the lawn of bacterial growth from each of three replicate plates was suspended in tryptic soy broth. Serial dilutions were plated on GCB-K to determine total CFUs per plate. CFU counts were normalized to the initial inoculum density (CFUs measured at time 0) for each replicate.

## Transmission electron microscopy

Bacterial cells grown overnight on GCB-K plates were suspended in liquid GCP medium with 1% Kellogg's supplement and 0.042% $NaHCO_3$ to $OD_{600}$ 0.1. Suspensions were incubated at 37°C with aeration for 2 hr to allow cultures to return to log phase. Cell pellets were collected by centrifugation and fixed for at least two hours at room temperature in 0.2M cacodylate buffer with 2.5% paraformaldehyde, 5% glutaraldehyde, and 0.06% picric acid. Fixed pellets were washed in 0.1M cacodylate buffer and postfixed with 1% Osmiumtetroxide ($OsO_4$)/1.5% Potassiumferrocyanide (KFeCN6) for 1 hr, washed 2x in water, 1x Maleate buffer (MB) 1x and incubated in 1% uranyl acetate in MB for 1 hr followed by two washes in water and subsequent dehydration in grades of alcohol (10 min each; 50%, 70%, 90%, 2 × 10 min 100%). The samples were then put in propyleneoxide for 1 hr and infiltrated ON in a 1:1 mixture of propyleneoxide and TAAB Epon (Marivac Canada Inc St. Laurent, Canada). The following day the samples were embedded in TAAB Epon and polymerized at 60 degrees C for 48 hr. Ultrathin sections (~60 nm) were cut on a Reichert Ultracut-S microtome, picked up on to copper grids stained with lead citrate and examined in a JEOL 1200EX Transmission electron microscope or a TecnaiG$^2$ Spirit BioTWIN and images were recorded with an AMT 2 k CCD camera.

## Transcriptomics

RNA isolation and sequencing was performed from *N. gonorrhoeae* strains as described *Wadsworth et al. (2018)*. Bacterial cells grown on GCB-K plates for 17 hr were suspended in liquid GCP medium with 1% IsoVitaleX and 0.042% $NaHCO_3$ to $OD_{600}$ 0.1. Suspensions were incubated at 37°C with aeration for 2–3 hr to allow cultures to return to log phase. Cells were collected to measure baseline transcriptional profiles. For samples measuring the effect of drug exposure, 0.008 μg/mL CRO was added and cultures were incubated at 37°C for an additional 90 min before cell collection. RNA was purified with the Direct-Zol kit (Zymo). Transcriptome libraries were prepared at the Broad Institute at the Microbial 'Omics Core using a modified version of the RNAtag-seq protocol (*Shishkin et al., 2015*). Five hundred nanograms of total RNA was fragmented, depleted of genomic DNA, dephosphorylated, and ligated to DNA adapters carrying 5'-AN8-3' barcodes of known sequence with a 5' phosphate and a 3' blocking group. Barcoded RNAs were pooled and depleted of rRNA using the RiboZero rRNA depletion kit (Epicentre). Pools of barcoded RNAs were converted to Illumina cDNA libraries in two steps: (i) reverse transcription of the RNA using a primer designed to the constant region of the barcoded adapter with addition of an adapter to the 3' end of the cDNA by template switching using SMARTScribe (Clontech) as described previously (*Zhu et al., 2001*); (ii) PCR amplification using primers whose 5' ends target the constant regions of the 3' or 5' adapter and whose 3' ends contain the full Illumina P5 or P7 sequences. cDNA libraries were sequenced on the Illumina Nextseq 500 platform to generate 50 bp paired-end reads. Barcode sequences were removed, and reads were aligned to the FA1090 reference genome. Differential expression analysis was conducted in DESeq2 v.1.16.1 (*Love et al., 2014*).

## Sequence data availability

Genomic and transcriptomic read libraries are available from the NCBI SRA database (accession number PRJNA540288).

## PBP abundance measurement

Protein abundance of PBP1, PBP2, and PBP3 was calculated using the fluorescent penicillin derivative bocillin-FL (Thermo Fisher). *N. gonorrhoeae* strains from overnight cultures were suspended in liquid GCP medium supplemented with 1% IsoVitalex and 0.042% $NaHCO_3$ to a density of $OD_{600}$ 0.1. Suspensions were incubated with aeration at 37°C for 2.5–3 hr. Bacterial cells were collected by centrifugation, washed once with 1 mL of sterile phosphate-buffered saline (PBS), and resuspended in PBS 5 μg/mL bocillin-FL and 0.1% dimethyl sulfoxide (DMSO) to a final concentration of 1 mL of $OD_{600}$ 0.5 per 50 μL suspension. Bocillin-FL suspensions were incubated for 5 min. An equal volume 2x SDS-PAGE sample buffer (Novex) was added and samples were boiled for 5 min. Proteins in 30 μL of each suspension were separated by SDS-PAGE on 4–12% Tris-Glycine protein gels (Novex), which were visualized on a Typhoon imager (Amersham) (excitation 488 nm/emission 526 nm) to detect bocillin-FL fluorescence. Densitometry was performed with ImageJ (*Schneider et al., 2012*).

Total bocillin-FL fluorescent signal was calculated for each sample, and the proportional contribution of individual PBPs to this signal was reported.

## Muropeptide analysis

*N. gonorrhoeae* strains were cultured for approximately 18 hr at 37°C with agitation in GCP medium supplemented with Kellogg's reagent and 0.042% NaHCO$_3$. Peptidoglycan was isolated and digested as described *Kühner et al. (2015)* with minor modifications. Briefly, bacterial cells were pelleted by centrifugation, suspended in 1 mL 1M NaCl, and incubated at 100°C for 20 min. Samples were centrifuged for one minute at 18,000 $\times$ *g*. Pellets were washed three times with water, suspended in 1 mL water, and placed in a bath sonicator for 30 min. 0.5 mL 0.1M Tris pH 6.8, 40 μg/mL RNase, and 16 μg/mL DNase were added to each sample; samples were incubated with shaking at 37°C for two hours, with the addition of 16 μg/mL trypsin after the first hour of incubation. Samples were heated to 100°C for 5 min to inactivate enzymes, then centrifuged for 3 min at 18,000 $\times$ *g* to pellet peptidoglycan. Pellets were washed with 1 mL aliquots of water until the suspension pH measured between 5 and 5.5. Peptidoglycan was then resuspended in 0.2 mL 12.5 mM NaH$_2$PO$_4$ pH 5.5 with 500 U/mL mutanolysin and incubated with shaking at 37°C for 16 hr. Samples were heated to 100°C for 5 min to inactivate mutanolysin and centrifuged for 5 min at 18,000 $\times$ *g* to pellet debris. The supernatant, containing solubilized muropeptides, was removed to new tubes, and 50 μL 10 mg/mL NaBH$_4$ was added to each. Samples were incubated at room temperature for 30 min and then the pH was adjusted to 2–3 with 85% H$_3$PO$_4$.

LC-MS was conducted using an Agilent Technologies 1200 series HPLC in line with an Agilent 6520 Q-TOF mass spectrometer operating with electrospray ionization (ESI) and in positive ion mode. The muropeptide fragments were separated on a Waters Symmetry Shield RP18 column (5 μm, 4.6 $\times$ 250 mm) using the following method: 0.5 mL/minute solvent A (water, 0.1% formic acid) for 10 min followed by a linear gradient of 0% solvent B (acetonitrile, 0.1% formic acid) to 20% B over 90 min.

## Acknowledgements

We thank the Centers for Disease Control and Prevention Gonococcal Isolate Surveillance Project, Joseph Dillard, Steven Johnson, and Caroline Genco for generously providing strains; Georgia Lagoudas and Paul Blainey for sequencing undirected transformants; Jessica Alexander, Jonathan Livny, and the Broad Institute Microbial 'Omics Core for RNAseq support; the Harvard Medical School Electron Microscopy Facility for TEM imaging; and Crista Wadsworth and Mohamad Rustom Abdul Sater for advising on analysis pipelines.

This work was supported by the Richard and Susan Smith Family Foundation (to YHG) and the National Institutes of Health R01 AI132606 (to YHG), R01 GM76710 (to SW), T32 GM007753 (to DHFR), and F32 GM123579 (to MAW).

## Additional information

### Funding

| Funder | Grant reference number | Author |
|---|---|---|
| Richard and Susan Smith Family Foundation | | Yonatan H Grad |
| National Institutes of Health | R01 AI132606 | Yonatan H Grad |
| National Institutes of Health | R01 GM76710 | Suzanne Walker |
| National Institutes of Health | F32 GM123579 | Michael A Welsh |
| National Institutes of Health | T32 GM007753 | Daniel H F Rubin |

The funders had no role in study design, data collection and interpretation, or the decision to submit the work for publication.

## Author contributions

Samantha G Palace, Conceptualization, Data curation, Formal analysis, Validation, Investigation, Visualization, Methodology, Writing - original draft, Writing - review and editing; Yi Wang, Formal analysis, Investigation, Writing - review and editing; Daniel HF Rubin, Formal analysis, Validation, Investigation, Visualization, Writing - review and editing; Michael A Welsh, Investigation, Methodology, Writing - review and editing; Tatum D Mortimer, Data curation, Investigation, Visualization, Writing - review and editing; Kevin Cole, Investigation, Writing - review and editing; David W Eyre, Resources, Supervision, Funding acquisition, Writing - review and editing; Suzanne Walker, Resources, Supervision, Funding acquisition, Investigation, Writing - review and editing; Yonatan H Grad, Conceptualization, Resources, Supervision, Funding acquisition, Writing - review and editing

## Author ORCIDs

Samantha G Palace (iD) https://orcid.org/0000-0002-7849-8078
Michael A Welsh (iD) http://orcid.org/0000-0001-8268-6285
Yonatan H Grad (iD) https://orcid.org/0000-0001-5646-1314

## Decision letter and Author response

Decision letter https://doi.org/10.7554/eLife.51407.sa1
Author response https://doi.org/10.7554/eLife.51407.sa2

---

# Additional files

## Supplementary files

• Supplementary file 1. Allelic diversity of RNA polymerase holoenzyme components and sigma factors.

• Supplementary file 2. Ceftriaxone susceptibility and genotype of isolates that have or can acquire CRO$^{RS}$-associated RNA polymerase mutation.

• Supplementary file 3. Muropeptide masses detected in cell wall digests of GCGS0457 and RNAP mutants.

• Supplementary file 4. *N. gonorrhoeae* strains used in this study.

• Transparent reporting form

## Data availability

Sequencing data have been deposited in the NCBI SRA database under accession number PRJNA540288.

The following dataset was generated:

| Author(s) | Year | Dataset title | Dataset URL | Database and Identifier |
|---|---|---|---|---|
| Palace SG, Wang Y, Grad YH | 2019 | N. gonorrhoeae isolates with reduced ceftriaxone susceptibility | https://www.ncbi.nlm.nih.gov/bioproject/PRJNA540288/ | NCBI BioProject, PRJNA540288 |

The following previously published datasets were used:

| Author(s) | Year | Dataset title | Dataset URL | Database and Identifier |
|---|---|---|---|---|
| Grad YH, Kirkcaldy RD, Trees D, Dordel J, Harris SR, Goldstein E, Weinstock H, Parkhill J, Hanage WP, Bentley S, Lipsitch M | 2013 | Emergence and spread of antibiotic resistant Neisseria gonorrhoeae in the US | https://www.ebi.ac.uk/ena/data/view/PRJEB2999 | ENA, PRJEB2999 |
| Grad YH, Harris SR, Kirkcaldy RD, Green AG, Marks DS, Bentley SD, | 2016 | Extended_genomic_epidemiology_of_Neisseria_gonorrhoeae_with_reduced_susceptibility_to_extended_ | https://www.ebi.ac.uk/ena/data/view/PRJEB7904 | ENA, PRJEB7904 and PRJEB2090 |

| | | | | |
|---|---|---|---|---|
| Trees D, Lipsitch M | | spectrum_cephalosporins_and_azithromycin_in_the_US | | |
| De Silva D, Peters J, Cole K, Cole MJ, Cresswell F, Dean G, Dave J, Thomas D Rh, Foster K, Waldram A, Wilson DJ, Didelot X, Grad YH, Crook DW, Peto TEA, Walker AS, Paul J | 2016 | Whole-genome sequencing to determine Neisseria gonorrhoeae transmission: an observational study. | https://www.ncbi.nlm.nih.gov/sra/?term=SRR3360905 | NCBI Sequence Read Archive, SRR3360905 |

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
