## [Decision Letter]

Thank you for submitting your article "RNA polymerase mutations cause cephalosporin resistance in clinical Neisseria gonorrhoeae isolates" for consideration by *eLife*. Your article has been reviewed by three peer reviewers, and the evaluation has been overseen by a Reviewing Editor and Gisela Storz as the Senior Editor. The following individual involved in review of your submission has agreed to reveal their identity: William Shafer (Reviewer #1).

The reviewers have discussed the reviews with one another and the Reviewing Editor has drafted this decision to help you prepare a revised submission.

Summary:

This manuscript reports mutations in RNA polymerase (RNAP) that confer elevated resistance to ceftriaxone in *N. gonorrhoeae*. The authors propose a mechanism whereby the mutations in RNAP lead to over-expression of PBP1, which contributes to the ceftriaxone/cefixime resistance. These findings are significant as they reveal a new mechanism for ceftriaxone resistance that could impact the problem of cephalosporin resistance.

Essential revisions:

1) Have the authors introduced the *rpoB* or *rpoD* mutations into isogeneic strains that differ only in the presence of a wild-type or mutant *ponA* gene? It strikes me that all of the clinical isolates have the *ponA* mutation. The question is really whether the host strain has to have a mutant *ponA* or is just over-expression of *ponA* sufficient for *rpoB/D*-driven resistance. The same is true for the *mtrR* mutation since loss of MtrR activity may impact transcription of *ponA* and the *rpoB*/D mutations could compensate.

2) The authors indicate that the *rpoB/rpoD* mutations do not result in a growth defect. The methodology used and the description is difficult to follow as written. Did they perform a competitive growth experiment in which parent and transformant strains were co-cultivated in broth and then sampled over time to distinguish the two strains? In any event, it would be good to present a standard growth curve with OD and CFU/mL values.

3) If known, please state the spontaneous mutation rates for *rpoB/rpoD* mutations with respect to ceftrixone resistance. I ask this because in one case spontaneous mutants in transformation experiments were not obtained (although the number of bacteria may have been too low to detect). In the second case, spontaneous resistant mutants were obtained.

4) The claim of altered peptidoglycan structure in the presence of the RNAP mutations appears weak in the absence of statistical analysis of peaks (Figure 5), therefore there is concern about the rigor of these particular experiments. Only the abundance of pentapeptides changes and the difference is two-fold. The data shown are said to be representative of three independent experiments, and it should therefore be facile to derive errors for the abundance of PG species.

5) There are some discrepancies between Table 1 and the described written results. Subsection “A single missense mutation in *rpoB* is the genetic basis for reduced ceftriaxone susceptibility in the clinical isolate GCGS1095”, last paragraph. The MIC of the transformant GCGS0457 RpoB^R201H^ was not similar to the MIC of the parental strain GCGS1095 – it was 2x less. How many times were the MICs repeated and how much variation was observed? Were all MICs conducted using the same method or were some strains tested by agar dilution and others by Etest? I agree that the transformation resulted in a highly elevated CRO MIC but the descriptions should be exact.

6) The written text states that “GCGS0457 RpoD^E98K^ had a CRO MIC of 0.125” while Table 1 indicates an MIC of 0.25. This is not similar to the parental strain.

7) Figure 1. A better description of the database used to generate this tree would be helpful. Perhaps this could be included in the Materials and methods section. The tree should be enlarged for clarity, including the insert. The dots on the phylogenetic tree of the insert should be connected with relevant strain numbers for easier interpretation. How many GISP isolates does this tree include? The third paragraph of the subsection “RNA polymerase mutations explain CRO^RS^ in other clinical isolates” mentions 1102 genomes; this could be included in the figure and in a better description of the GISP data base in the Materials and methods.

8) Supplementary file 1 should be reformatted to identify variants, possibly by bolding, coloring, or separating the variants to distinguish them.

9) Section on clinical isolates from multiple lineages. There is no information on what isolates were transformed – the panel of 17 should be named (Materials and methods section) and the exact transformation procedure described – was the PCR product used for transformation? Is a onetime (subsection “Clinical isolates from multiple lineages can achieve high-level CRO^RS^ via a single nucleotide change in *rpoB*”, last paragraph) transformation procedure really indicative of a failed transformation? These descriptions are quite incomplete, and could be described briefly but succinctly. Presumably, some of these strains are indicated in Figure 1, although the figure is difficult to read.

10) Could it be that the mutants may have prolonged stationary phase survival (Figure 3)?

11) The authors indicate that the transcription of *mtrCDE* was not elevated. It appears that transcription was uniformly decreased in mutant strains. Is this decrease significant? What would/do these results imply?

---

## [Author Response]

Essential revisions:1) Have the authors introduced the rpoB or rpoD mutations into isogeneic strains that differ only in the presence of a wild-type or mutant ponA gene? It strikes me that all of the clinical isolates have the ponA mutation. The question is really whether the host strain has to have a mutant ponA or is just over-expression of ponA sufficient for rpoB/D-driven resistance. The same is true for the mtrR mutation since loss of MtrR activity may impact transcription of ponA and the rpoB/D mutations could compensate.

We have now updated the manuscript to address these points. First, with regard to PonA: our transformation experiments showed that the PBP1 L421P variant is not required for RNAP-mediated CRO^RS^. Among the panel of isolates that have been successfully transformed to a CRO^RS^ phenotype via *rpoB1*, three isolates – GCGS0275, GCGS0465, and GCGS0524 – lack this *ponA* variant. This information is available in the new Supplementary file 2 and discussed in the expanded transformation section that has been included in response to point 9 (below). Although we have not constructed an isogenic strain pair to directly determine the effects of the PBP1 L421P mutation, we believe the data above support that this variant is not required.

Second, with regard to MtrR: similar to PonA, the variant allele of *mtrR* that is present in the parental CRO^RS^ isolates GCGS1013, GCGS1014, GCGS1095, and GCGS0364 is not required for RNAP-mediated reduced CRO susceptibility. Several *mtrR* alleles are represented among the panel of isolates that were successfully transformed with *rpoB1*, including the wildtype allele in isolate GCGS0275. This information is included in Supplementary file 2.

2) The authors indicate that the rpoB/rpoD mutations do not result in a growth defect. The methodology used and the description is difficult to follow as written. Did they perform a competitive growth experiment in which parent and transformant strains were co-cultivated in broth and then sampled over time to distinguish the two strains? In any event, it would be good to present a standard growth curve with OD and CFU/mL values.

In preliminary growth curve experiments, we found that tracking growth of cultures growing in liquid GCP media supplemented with 1% IsoVitaleX and 0.042% NaHCO_3_ by OD_600_ measurements did not show the same time-dependent trends as tracking growth of these cultures by dilution and CFU plating. Similar results have been reported before, as in Figure 1 of Vincent et al. (2018, mBio, e01905-17).

For completeness, we have performed the suggested experiment. Briefly, the liquid cultures of GCGS0457 and its CRO^RS^ transformants GCGS0457 *rpoD1* and GCGS0457 *rpoB1* were seeded in GCP supplemented with 1% IsoVitaleX and 0.042% NaHCO_3_ at a starting density of approximately OD_600_ = 0.1. Culture density was monitored by spectrometry and dilution plating every 2 hours. Here we show mean and standard deviation of 3 technical replicates; the data are representative of 2 independent experiments.

The purpose of this growth curve in our work is not to argue that the RNAP mutations have no effect on fitness; indeed, given the broad perturbations we observe in transcriptional data, we find it likely that these RNAP mutants might be profoundly growth disadvantaged in many physiologically relevant conditions. Instead, we wished to determine whether the elevated MIC phenotype we observe in the antimicrobial sensitivity testing we report here might be related to a slow-growth phenotype. Therefore, we measured the growth of our strains of interest as they were growing on GCB agar plates, as this is the growth condition in which MIC measurements were performed, and therefore the condition most closely suited to determining how growth rate differences might affect results of the MIC assay.

3) If known, please state the spontaneous mutation rates for rpoB/rpoD mutations with respect to ceftrixone resistance. I ask this because in one case spontaneous mutants in transformation experiments were not obtained (although the number of bacteria may have been too low to detect). In the second case, spontaneous resistant mutants were obtained.

We have not measured the rate of spontaneous RNAP CRO^RS^ mutation. All selection for de novo or transformed CRO^RS^ RNAP mutants in this work was performed by plating bacterial suspensions on a plate containing a concentrated spot of CRO and identifying colonies growing within the zone of inhibition. This method improves the recovery CRO^RS^ mutants, which often appear close to the margin of the zone of inhibition; we have found that plating efficiency of CRO^RS^ strains on even modest CRO concentrations is poor, perhaps explaining this pattern. However, as this method does not permit the robust quantification of the number of CRO^RS^ CFUs over the entire surface of a plate, it is not compatible with traditional methods for measuring mutation or transformation rates.

4) The claim of altered peptidoglycan structure in the presence of the RNAP mutations appears weak in the absence of statistical analysis of peaks (Figure 5), therefore there is concern about the rigor of these particular experiments. Only the abundance of pentapeptides changes and the difference is two-fold. The data shown are said to be representative of three independent experiments, and it should therefore be facile to derive errors for the abundance of PG species.

We appreciate this opportunity to make the case more compellingly, and we have now included statistical information in Figure 5, as well as data from individual experiments in Figure 5—source data 1.

5) There are some discrepancies between Table 1 and the described written results. Subsection “A single missense mutation in rpoB is the genetic basis for reduced ceftriaxone susceptibility in the clinical isolate GCGS1095”, last paragraph. The MIC of the transformant GCGS0457 RpoB^R201H^ was not similar to the MIC of the parental strain GCGS1095 – it was 2x less. How many times were the MICs repeated and how much variation was observed? Were all MICs conducted using the same method or were some strains tested by agar dilution and others by Etest? I agree that the transformation resulted in a highly elevated CRO MIC but the descriptions should be exact.

In general, two-fold variation in MIC determination is considered within the boundaries of technical variation – particularly when MIC testing is done by agar dilution, which determines MIC only to the nearest doubling dilution. The language in this section was chosen with this in mind.

In the original manuscript, reported MICs were a mixture of MICs determined by agar dilution, by Etest, and by agar dilution from the Centers for Disease Control and Prevention’s original GISP report (see Grad et al., 2016). We thank the reviewers for their suggestion that we improve the clarity of this section. For consistency, we are now showing MIC data that was collected either by Etest only or by agar dilution only, depending on the drug being tested (see Materials and methods, subsection “bacterial strains and culture conditions”). This required additional MIC testing to replace some MIC values that had previously been determined by agar dilution with new MIC values determined by Etest. MIC values in Tables 1 and 2 and Supplementary file 2 have been altered to reflect this more recent Etest data, which does not qualitatively change any of the interpretations of our paper, but which allows MIC determination at a finer resolution than agar dilution.

6) The written text states that “GCGS0457 RpoD^E98K^ had a CRO MIC of 0.125” while Table 1 indicates an MIC of 0.25. This is not similar to the parental strain.

This discrepancy was an error, originating from reporting the results of different replicate experiments in these two places (see point 5 above). We now report the redone Etest results, both in the text and in Tables 1 and 2.

7) Figure 1. A better description of the data base used to generate this tree would be helpful. Perhaps this could be included in the Materials and methods section. The tree should be enlarged for clarity, including the insert. The dots on the phylogenetic tree of the insert should be connected with relevant strain numbers for easier interpretation. How many GISP isolates does this tree include? The third paragraph of the subsection “RNA polymerase mutations explain CRO^RS^ 105 in other clinical isolates” mentions 1102 genomes; this could be included in the figure and in a better description of the GISP data base in the Materials and methods.

Figure 1 has been enlarged, and we have added additional annotations.

This phylogeny includes the 1102 genomes from the GISP project (Grad et al., 2016), plus the GCPH44 isolate identified from the De Silva et al., 2016 report. The figure legend has been updated to include this information. Metadata for the isolates in the GISP strain collection is available in the Grad et al., 2016 reference; the Materials and methods section has been revised to make this clear.

8) Supplementary file 1 should be reformatted to identify variants, possibly by bolding, coloring, or separating the variants to distinguish them.

Variants in this file, Supplementary file 1, have been bolded for emphasis. We also note that, if opened in Excel, this file is sortable by gene variant.

9) Section on clinical isolates from multiple lineages. There is no information on what isolates were transformed – the panel of 17 should be named (Materials and methods section) and the exact transformation procedure described – was the PCR product used for transformation? Is a onetime (subsection “Clinical isolates from multiple lineages can achieve high-level CRO^RS^ via a single nucleotide change in rpoB”, last paragraph) transformation procedure really indicative of a failed transformation? These descriptions are quite incomplete, and could be described briefly but succinctly. Presumably, some of these strains are indicated in Figure 1, although the figure is difficult to read.

We have expanded the description of this experiment in the text, listed the 17 isolates used for this panel in the Materials and methods section, included the 5 successful isolates in Figure 1, and added a table (Supplementary file 2) giving more detail about these isolates.

We agree that a single failed transformation experiment into any given clinical isolate does not represent strong evidence for genetic incompatibility and we have updated the text to reflect this.

10) Could it be that the mutants may have prolonged stationary phase survival (Figure 3)?

The growth curve data in Figure 3 suggests that these strains may have altered growth or survival phenotypes in stationary phase, and we look forward to exploring this possibility in follow-up work to more thoroughly characterize the effects of CRO^RS^-associated RNAP mutation on bacterial physiology. As the purpose of the growth curve experiment in the context of this study was to rule out slow growth as an explanation for increased growth of RNAP mutants on ceftriaxone MIC plates, we have not focused on the stationary phase portion of our growth curve, as differences in survival at this point would likely not affect MIC interpretation.

11) The authors indicate that the transcription of mtrCDE was not elevated. It appears that transcription was uniformly decreased in mutant strains. Is this decrease significant? What would/do these results imply?

This decrease is significant for *mtrC* and *mtrE*, as well as for *mtrD* in all cases except for the comparison of the GCGS0457 *rpoB1* transformant with GCGS0457, in which this decrease failed to meet the significance threshold (Benjamini-Hochberg adjusted p value 0.06 for this comparison). Interestingly, the decrease in transcription through the *mtrCDE* operon we observe occurs in the presence of a similar magnitude decrease in transcription of the *mtrR* repressor; the change we see in the *mtr* regulon is not coherent. Perhaps this is evidence for a general derangement or perturbation of transcriptional regulation broadly in RNAP mutants. In the context of RNAP-mediated CRO^RS^, the apparently decreased transcription of *mtrCDE* in the RNAP mutants is evidence that the concomitant CRO^RS^ phenotype is not due to increased drug efflux via the Mtr pump.